# DEEP ENSEMBLES FOR LOW-DATA TRANSFER LEARNING

## ABSTRACT

In the low-data regime, it is difficult to train good supervised models from scratch. Instead practitioners turn to pre-trained models, leveraging transfer learning. Ensembling is an empirically and theoretically appealing way to construct powerful predictive models, but the predominant approach of training multiple deep networks with different random initialisations collides with the need for transfer via pre-trained weights. In this work, we study different ways of creating ensembles from pre-trained models. We show that the nature of pre-training itself is a performant source of diversity, and propose a practical algorithm that efficiently identifies a subset of pre-trained models for any downstream dataset. The approach is simple: Use nearest-neighbour accuracy to rank pre-trained models, fine-tune the best ones with a small hyperparameter sweep, and greedily construct an ensemble to minimise validation cross-entropy. When evaluated together with strong baselines on 19 different downstream tasks (the Visual Task Adaptation Benchmark), this achieves state-of-the-art performance at a much lower inference budget, even when selecting from over 2,000 pre-trained models. We also assess our ensembles on ImageNet variants and show improved robustness to distribution shift.

## 1 INTRODUCTION

There are many ways to construct models with minimal data. It has been shown that fine-tuning pre-trained deep models is a compellingly simple and performant approach (Dhillon et al., 2020; Kolesnikov et al., 2019), and this is the paradigm our work operates in. It is common to use networks pre-trained on ImageNet (Deng et al., 2009), but recent works show considerable improvements by careful, task-specific pre-trained model selection (Ngiam et al., 2018; Puigcerver et al., 2020).

Ensembling multiple models is a powerful idea that often leads to better predictive performance. Its secret relies on combining different predictions. The source of diversity for deep networks has been studied (Fort et al., 2019; Wenzel et al., 2020), though not thoroughly in the low-data regime. Two of the most common approaches involve training independent models from scratch with (a) different random initialisations, (b) different random subsets of the training data. Neither of these are directly applicable downstream with minimal data, as we require a pre-trained initialisation to train competitive models[1], and data scarcity makes further data fragmentation impractical. We study some ways of encouraging model diversity in a supervised transfer-learning setup, but fundamentally argue that the nature of pre-training is itself an easily accessible and valuable form of diversity.

Previous works consider the construction of ensembles from a set of candidate models (Caruana et al., 2004). Services such as Tensorflow Hub (Google, 2018) and PyTorch Hub (FAIR, 2019) contain hundreds of pre-trained models for computer vision; these could all be fine-tuned on a new task to generate candidates. Factoring in the cost of hyperparameter search, this may be prohibitively expensive. We would like to know how suited a pre-trained model is for our given task *before* training it. This need has given rise to cheap proxy metrics which assess this suitability (Puigcerver et al., 2020). We use such metrics - leave-one-out nearest-neighbour ($k$NN) accuracy, in particular - as a way of selecting a *subset* of pre-trained models, suitable for creating diverse ensembles of task-specific experts. We show that our approach is capable of quickly narrowing large pools (up to 2,000) of candidate pre-trained models down to manageable (15 models) task-specific sets, yielding a practical algorithm in the common context of the availability of many pre-trained models.

---

[1]For an illustration of the importance of using pre-trained models in the low-data regime see Appendix C.1.

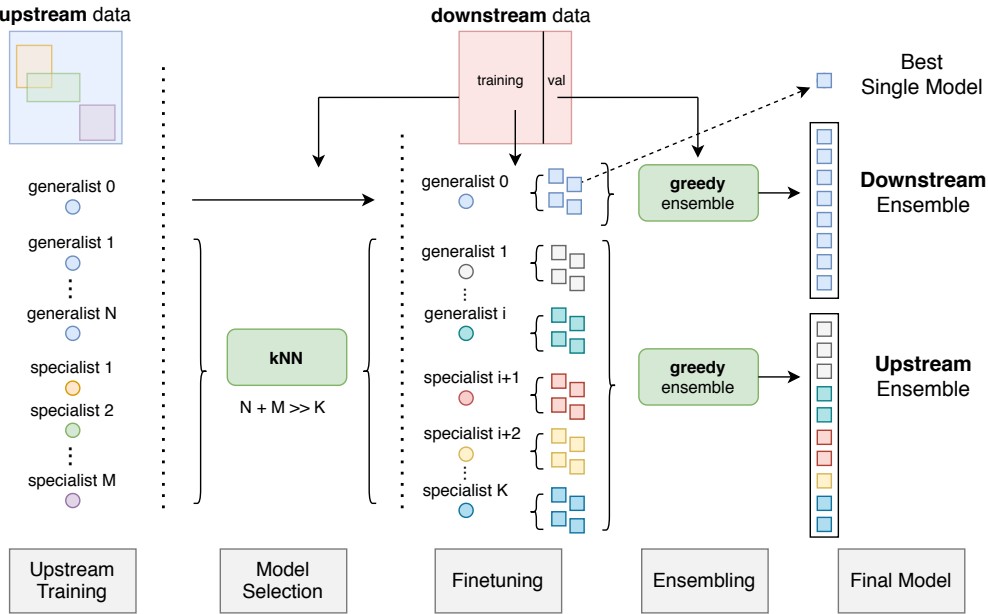

Figure 1: Overview of the different ways of constructing diverse ensembles studied in this work. We propose an algorithm that exploits diversity in a large pool of pre-trained models, by using leave-one-out $k$-nearest-neighbour ($k$NN) accuracy to select a subset to form the ensemble.

We first experiment with sources of downstream diversity (induced only by hyperparameterisation, augmentation or random data ordering), giving significant performance boosts over single models. Using our algorithm on different pools of candidate pre-trained models, we show that various forms of upstream diversity produce ensembles that are more accurate and robust to domain shift than this. Figure 1 illustrates the different approaches studied in our work. Ultimately, this new form of diversity improves on the Visual Task Adaptation Benchmark (Zhai et al., 2019) SOTA by 1.8%.

The contributions of this paper can be summarized as follows:

- We study ensembling in the context of transfer learning in the low data regime & propose a number of ways to induce advantageous ensemble diversity which best leverage pre-trained models.

- We show that diversity from upstream pre-training achieves better accuracy than that from the downstream fine-tuning stage (+1.2 absolute points on average across the 19 downstream classification VTAB tasks), and that it is more robust to distribution shift (+2.2 absolute average accuracy increase on distribution shifted ImageNet variants).

- We show that they also surpass the accuracy of large SOTA models (76.2% vs. 77.6%) at a much lower inference cost, and achieve equal performance with less than a sixth of the FLOPS.

- We extend the work from Puigcerver et al. (2020) and demonstrate the efficacy of $k$NN accuracy as a cheap proxy metric for selecting a *subset* of candidate pre-trained models.

## 2 CREATING ENSEMBLES FROM PRE-TRAINED MODELS

We first formally introduce the technical problem we address in this paper. Next we discuss baseline approaches which use a single pre-trained model, and then we present our method that exploits using multiple pre-trained models as a source of diversity.

### 2.1 THE LEARNING SETUP: UPSTREAM, MODEL SELECTION, DOWNSTREAM

Transfer learning studies how models trained in one context boost learning in a different one. The most common approach pre-trains a single model on a large dataset such as ImageNet, to then tune the model weights to a downstream task. Despite algorithmic simplicity, this idea has been very

successful. In a downstream low-data scenario, it is more difficult for a one-size-fits-all approach to triumph as specializing the initial representation becomes harder. As in Puigcerver et al. (2020), we explore the scenario where a *range* of pre-trained models is available, and we can look at the target data to make a decision on which models to fine-tune. However, we generalize and improve it by simultaneously selected several models for fine-tuning, since downstream tasks may benefit from combining expert representations aimed at capturing different aspects of the learning task: for instance, on a natural scenery dataset one could merge different models that focus on animals, plants, food, or buildings. Fine-tuning all pre-trained models to pick the best one is a sensible strategy, but rarely feasible. To keep the algorithms practical, we identify two compute budgets that should be controlled for: The *fine-tuning* budget, i.e. the total number of models we can fine-tune on a downstream task; and the *inference* budget, the maximum size of the final model.

## 2.2 BASELINES: DIVERSITY FROM DOWNSTREAM TRAINING

The baselines we propose leverage transfer learning by requiring a pre-trained model - this is crucial, see Appendix C.1. We use a strong *generalist* model (BiT-ResNet 50s from Kolesnikov et al. (2019), trained on all upstream data) and consider three methods to create a model set for ensemble selection.

**Random Seeds**. Fine-tuning a generalist model multiple times with *fixed* hyperparameters will yield different classifiers, analogous to the DeepEnsembles of Lakshminarayanan et al. (2017). Note, here we can only take advantage of randomised data ordering/augmentation, which Fort et al. (2019) showed, though useful, was not as beneficial as diversity from random initalisation.

**HyperEnsembles**. Hyperparameter diversity was recently shown to further improve DeepEnsembles (Wenzel et al., 2020). We define a hyperparameter search space, randomly sample as many configurations as we have fine-tuning budget, and fine-tune the generalist on downstream data with each of those configurations. Further details on training are given in Appendix A.2.

**AugEnsembles**. We generate a set of models by fine-tuning the generalist on each task with randomly sampled *families* of augmentation (but fixed hyperparameters). Details are in Appendix A.3.

## 2.3 OUR METHOD: DIVERSITY FROM UPSTREAM PRE-TRAINING

Fort et al. (2019) explain the strong performance of classical ensembling approaches – independently training randomly initialised deep networks – by showing that each constituent model explores a different mode in the function space. For transfer learning, Neyshabur et al. (2020) show that with pre-trained weights, fine-tuned models stay in a local 'basin' in the loss landscape. Combining both gives a compelling reasoning for the use of multiple pre-trained networks for transfer with ensembles, as we propose here. Instead of diversity from downstream fine-tuning, we show that in the low data regime, better ensembles can be created using diversity from pre-training.

We consider three sources of upstream diversity. First, we consider generalists that were *pre-trained* with different random seeds on the same architecture and data. Second, we consider *experts*, specialist models which were pre-trained on different subsets of the large upstream dataset. Lastly, we exploit diversity in scale – pre-trained models with architectures of different sizes. Given a pool of candidate models containing such diversity, we propose the following algorithm (Figure 1):

**1. Pre-trained model selection.** Fine-tuning all experts on the new task would be prohibitively expensive. Following Puigcerver et al. (2020), we rank all the models by their $k$NN leave-one-out accuracy as a proxy for final fine-tuned accuracy, instead keeping the $K$ best models (rather than 1).

**2. Fine-tuning.** We add a fully connected layer to each model's final representation, and then train the whole model by minimising categorical cross-entropy via SGD. Given a pool of $K$ pre-trained models from stage 1, we tune each with 4 learning rate schedules, yielding a total of $L = 4K$ models for the step 3 (Usually $K = 15$ and $L = 60$). See Appendix A.1.1 for more details.

**3. Ensemble construction.** This is shared among all presented ensembles. We use the greedy algorithm introduced by Caruana et al. (2004). At each step, we greedily pick the next model which minimises cross-entropy on the validation set when it is ensembled with already chosen models.

These steps are independently applied to each task; each step makes use of the downstream dataset, so each dataset gets a tailored set of pre-trained models to create the ensemble pool and therefore

very different final ensembles result. We also considered a greedy ensembling algorithm in $k$NN space which aims to sequentially pick complementary models which will likely ensemble well together (Appendix C.6), but picking top-$K$ was generally better.

### 2.3.1 COMBINED APPROACHES

The diversity induced by different upstream models and distinct downstream hyperparameters should be complementary. Given a fine-tuning budget of $L$, we can set the number of pre-trained models $K$ in advance, providing each of them with a random hyperparameter sweep of size $L/K$. However, for some tasks it may be more beneficial to have fewer different pre-trained models and a wider sweep, or vice versa. We aim to dynamically set this balance per-dataset using a heuristic based on the $k$NN accuracies; namely, we keep all pre-trained models within some threshold percentage $\tau\%$ of the top $k$NN accuracy, up to a maximum of $K = 15$. Ideally, this would adaptively discard experts poorly suited to a given task, whose inclusion would likely harm ensemble performance. The saved compute budget is then used to squeeze more performance from available experts by testing more hyperparameters, and hopefully leading to greater *useful* diversity. We arbitrarily set $\tau = 15\%$ for our experiments, but this choice could likely be improved upon. Appendix C.5 shows how the number of models picked varies per task, and the gains with respect to having a fixed $K$.

### 2.3.2 PRE-TRAINING MODELS

We use BiT ResNets pre-trained on two large upstream datasets with hierarchical label spaces: JFT-300M (Sun et al., 2017) and ImageNet-21k (Deng et al., 2009). We consider two types of pre-trained models. *Generalists* are trained on the entire upstream dataset. In particular, we consider 15 JFT ResNet-50 generalists that were pre-trained with different random initalisations. *Experts* are generated by splitting the hierarchical label spaces into sub-trees and training independent models on the examples in each sub-tree. We pre-train 244 experts from JFT and 50 from ImageNet21k, following the protocol of (Puigcerver et al., 2020) (see Appendix A.1). For low-data downstream tasks, this is by far the most expensive stage of the process. It is however only incurred once, and its cost is amortized as new downstream tasks are served, since any downstream task can reuse them.

## 3 ENSEMBLE EVALUATION

**Downstream Tasks**. We evaluate our models on the Visual Task Adaptation Benchmark (Zhai et al., 2019): 19 diverse downstream classification tasks, split into 'natural', 'specialised' and 'structured' categories. As we are primarily interested in low-data regimes, the tasks only have 1000 training datapoints (i.e., $\text{VTAB}_{1K}$) with a number of classes ranging from 2 to 397. We split data into 800 training examples and 200 validation examples. See Appendix B.1 for more information.

**Test Performance**. For our final competitive models, we first train all the individual models on the 800 training points. Then, we use the 200 validation data points to find the best ensemble (both running the greedy algorithm and choosing the overall ensemble size). For the resultant ensemble, we retrain constituent models on the full 1000 data points, and evaluate it on the test data.

**Robustness**. We train ExpertEnsembles and HyperEnsembles on ImageNet (Deng et al., 2009). While ImageNet does not match our low-data regime of interest, previous work and additional datasets allow us to conveniently measure robustness and uncertainty metrics. Thus, alongside reporting the accuracy on the official validation split, we assess models on a number of ImageNet-based robustness benchmarks, aiming to quantify calibration and robustness to distribution shift (Djolonga et al., 2020). More details on these variants are available in Appendices B.2 and A.4.

## 4 EXPERIMENTAL RESULTS

Unless otherwise specified, all experiments use a fine-tuning budget and inference budget of 60 and 15 models respectively. This was set arbitrarily; we experiment with both budgets to see the effect.

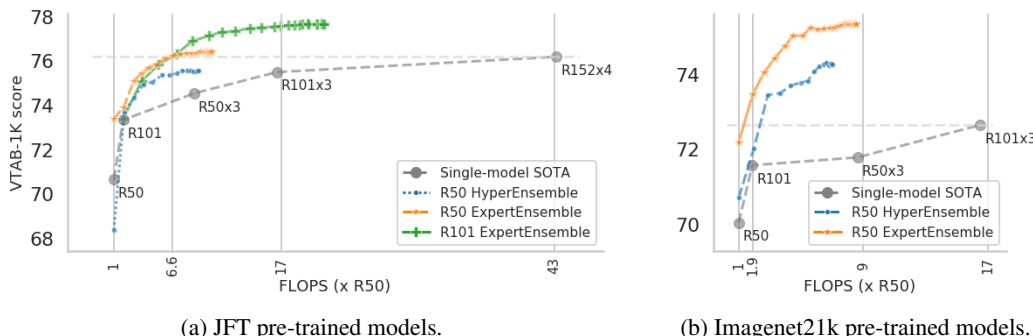

(a) JFT pre-trained models.        (b) Imagenet21k pre-trained models.

Figure 2: Inference cost vs. VTAB$_{1K}$ performance. State-of-the-art generalist models of different scales are compared against ensembles with varying inference budgets.

## 4.1 THE VALUE OF ENSEMBLES

We first show that ensembles in low-data transfer regimes dramatically beat their single-model counterparts –which are often much larger networks. Figure 2 and Table 1 compare our best ensembles (which all use upstream or combined diversity) on VTAB$_{1K}$ tasks. Our baselines are BiT models from (Kolesnikov et al., 2019), which had until now state-of-the-art performance.

**JFT pre-trained models.** The most standard approach –fine-tuning a single R50 model trained on all of JFT– leads to an average accuracy of 70.6%. It greatly lags behind compared to the ensemble-based algorithms; in particular, the difference is striking for structured and natural datasets. On average, the ensembles selected between 9 and 10 downstream models –this number greatly varies depending on the task, e.g. 3 were selected for CalTech101 and 14 for Diabetic Retinopathy. Accordingly, capacity-wise, it makes sense to compare the ensembles to larger ResNets. Table 1 shows that the JFT R50 ensembles match or slightly beat the performance of a R152x4. In particular, the ensembles offer a large advantage in settings where tasks diverge from single-object recognition, e.g. in the structured datasets. Even in natural domains, the experts have a better accuracy/FLOP ratio than the R152x4, which has $40\times$ more parameters than a single R50. Even more significant is the difference in inference *time*, as ensemble predictions can be easily parallelized.

**ImageNet21k pre-trained models.** The story is fairly similar for the pool of ImageNet21k experts. The ensembles select on average between 7 and 8 models –again, strongly task-dependent. Ensembles' average performance improvement with respect to a single R50 trained on all of ImageNet21k is around 5 absolute points (more than 7%). We also consider a much larger generalist baseline, in this case a R101x3, which has more than $16\times$ as many parameters as a single R50. Still the ensembles far outperform it, especially in structured datasets.

Table 1: Test accuracy of our best ensembles against reproduced baselines from Kolesnikov et al. (2019) [*]. For each dataset, we take the median of three independent runs. Rows show the average over datasets. Bootstrapped confidence intervals at the 95% level. The source of diversity for ensembles is shown: U = upstream (during pre-training) and C = combined (pre-training and fine-tuning).

| Description | Diversity | VTAB$_{1K}$ | Natural | Specialised | Structured |
|---|---|---|---|---|---|
| **JFT** BiT-L R50* | — | 70.6$_{[70.4 - 71.0]}$ | 77.8 | 83.6 | 57.9 |
| **JFT** BiT-L R152x4* | — | 76.2$_{[74.5 - 76.7]}$ | 86.0 | 87.0 | 62.2 |
| **JFT** R50 Experts + Generalists | U | 76.8$_{[76.4 - 77.0]}$ | 82.6 | 85.8 | 67.2 |
| **JFT** R101 Experts | U | **77.6**$_{[77.4 - 77.8]}$ | 83.6 | 86.4 | 68.0 |
| **INet21k** BiT-M R50* | — | 70.0$_{[69.6 - 70.5]}$ | 77.0 | 84.7 | 56.6 |
| **INet21k** BiT-M R101x3* | — | 72.7$_{[72.1 - 73.5]}$ | 80.3 | 85.7 | 59.4 |
| **INet21k** R50 Experts | U | 75.3$_{[74.5 - 75.6]}$ | 79.9 | 85.7 | 66.1 |
| **INet21k** R50 HyperExperts | C | **75.6**$_{[74.8 - 75.8]}$ | 79.9 | 85.5 | 67.0 |

Table 2: **Upstream diversity gives better ensembles**. Test accuracy of different ensembles. For each dataset, we take the median of three independent runs. Rows show the average over datasets. Bootstrapped confidence intervals at the 95% level. The source of diversity is noted as: D = downstream (during fine-tuning), U = upstream (during pre-training) and C = combined (both).

| Description | Diversity | VTAB$_{1K}$ | Natural | Specialised | Structured |
|---|---|---|---|---|---|
| **JFT** R50 Seeds | D | 74.9$_{[74.5 - 75.1]}$ | 78.5 | 85.9 | 66.2 |
| **JFT** R50 Augs | D | 73.7$_{[73.4 - 75.7]}$ | 80.6 | 86.8 | 61.2 |
| **JFT** R50 Hypers | D | 75.6$_{[75.2 - 75.7]}$ | 80.1 | 85.6 | 66.6 |
| **JFT** R50 Generalists+Experts | U | 76.8$_{[76.4 - 77.0]}$ | 82.6 | 85.8 | 67.2 |
| **JFT** R50 Aug(Gen. + Experts) | C | **77.6** $_{[76.8 - 77.9]}$ | 82.7 | 86.4 | 68.7 |
| **JFT** R50 Experts | U | 76.4$_{[76.1 - 76.6]}$ | 82.2 | 85.6 | 66.8 |
| **JFT** R50 HyperExperts | C | 76.6$_{[76.0 - 76.7]}$ | 82.5 | 85.8 | 66.8 |
| **JFT** R50 AugExperts | C | **76.8**$_{[76.6 - 76.9]}$ | 82.6 | 86.0 | 67.2 |
| **INet21k** R50 Hypers | D | 74.2$_{[73.7 - 74.8]}$ | 79.6 | 86.2 | 63.5 |
| **INet21k** R50 Experts | U | 75.3$_{[74.5 - 75.6]}$ | 79.9 | 85.7 | 66.1 |
| **INet21k** R50 HyperExperts | C | **75.6**$_{[74.8 - 75.8]}$ | 79.9 | 85.5 | 67.0 |

Table 3: **Ablations**. Test accuracy of different ensembles. For each dataset, we take the median of three independent runs. Rows show the average over datasets. Bootstrapped confidence intervals at the 95% level. Pre-training done on JFT, except for "All Experts" that also used ImageNet21k.

| | Description | VTAB$_{1K}$ | Natural | Specialised | Structured |
|---|---|---|---|---|---|
| Base | R50 Generalists+Experts | 76.8$_{[76.4 - 77.0]}$ | 82.6 | 85.8 | 67.2 |
| *Specialists vs* | R50 Experts | 76.4$_{[76.1 - 76.6]}$ | 82.2 | 85.6 | 66.8 |
| *generalists* | R50 Generalists | 76.5$_{[75.9 - 76.7]}$ | 81.3 | 86.2 | 67.7 |
| *Combine scales* | R18/34/50 Experts | 76.8$_{[76.4 - 77.2]}$ | 82.2 | 85.5 | 67.9 |
| *Stack with scale* | R101 Experts | 77.6$_{[77.4 - 77.8]}$ | 83.6 | 86.4 | 68.0 |
| *Massive pool* | All Experts (JFT/INet21k) | 77.6$_{[77.3 - 77.7]}$ | 83.6 | 86.1 | 68.1 |

**Overall.** A more complete story supporting the ensembles' value is depicted in Figure 2. The gray dashed line represents the previous Pareto frontier of VTAB$_{1K}$ average accuracy per FLOP. The ensemble models dominate the state-of-the-art, indicating their efficacy in the low-data regime.

## 4.2    THE VALUE OF UPSTREAM DIVERSITY

Results in Table 2 suggest that upstream diversity improves downstream diversity. For both JFT and ImageNet21k pre-training, ensembles benefiting from upstream sources of diversity outperform their downstream-based counterparts. Combining both gives a further small boost for expert ensembles.

Ablations on JFT pre-trained models are shown in Table 3; we now discuss the learnings from that.

*Experts help when pre-training is relevant*: Results on Table 2 used $k$NN to pick from a pool of 15 generalists and 244 experts. We break these pools down separately. Experts give a significant boost on Natural datasets; the ensembles take advantage of the relevance of experts pre-trained on the predominately 'natural' slices of the upstream data. For datasets without clear experts, there is less benefit to this approach, and the generalists shine.

*Performance improvements stack with scale*: The strong performance of the R101 Expert Ensemble shows performance gains stack somewhat with scale; it improves on R50 Experts by 1.2% absolute points in accuracy, improving in all categories. We explore this more thoroughly in Appendix C.7.

*Combining upstream and downstream diversity helps*: As discussed in Section 2.3.1, we combine experts with hyper or augmentation ensembles. The simple approach of thresholding by $k$NN accuracy works well, giving a small boost in test performance (R50 *Aug(Gen. + Experts)*, R50 *HyperExperts*, R50 *AugExperts*). More details on this are provided in Appendix C.5.

### 4.3 THE VALUE OF NEAREST NEIGHBOUR SELECTION

*kNN may possibly help*: The greedy ensemble algorithm is not perfect, and with such a small validation set it is prone to overfit. When *all* upstream JFT R50 experts are fine-tuned and passed to the greedy algorithm, test performance drops slightly. We further explore this in Appendices C.4, C.9.

*It can compare models of different sizes*: Overall, larger models perfom better at transfer (Kolesnikov et al., 2019). Per-dataset, this is not the case; e.g. we found R34 experts were best on structured tasks. One may expect $k$NN selection or the greedy algorithm to be biased towards selecting larger architectures. The final ensembles instead use a mix of scales. The R18/R34/R50 experts ensemble improves on just R50s by 0.4%, indicating possible benefits; more discussion is in Appendix C.7.

*Can filter a very large pool of models*: When selecting only 15 pre-trained models from over 2000 candidates (different architecture sizes and upstream datasets), the overall VTAB performance (*All Experts* in Table 3) is similar to only selecting from ResNet-101s. This highlights the remarkable robustness of our model selection. These results are broken down further in Appendix C.8.

Mirroring Puigcerver et al. (2020), we have shown $k$NN to be a cheap yet successful way of selecting models. It is not perfect - when combining pools, one would hope for at least a 'best of both' performance. $k$NN selection wasn't needed for generalists (we had 15 pre-trained models), but when combining the generalists and experts in a pool, specialised/structured performance drops slightly.

### 4.4 EFFECT OF FINE-TUNING BUDGET

In most experiments, the $k$NN picks $K = 15$ experts. With the default 4 hyperparameters, this is a fine-tuning budget of 60 models. This is the number of models trained for a given task, and the majority of compute expenditure incurred by a practitioner, as the $k$NN selection/ensembling are comparatively cheap. The hyperensemble was run with the same budget. Figure 3 shows how performance drops with reduced fine-tuning budget. Interestingly, the expert ensembles are actually more robust to a reduced budget, retaining higher performance when training fewer models, indicating the $k$NN's usefulness as a pre-selection phase.

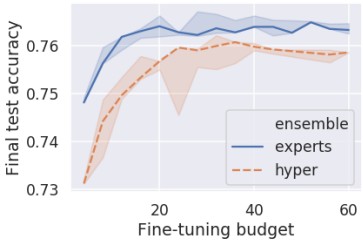

Figure 3: Effect of fine-tuning budget on ensemble VTAB$_{1K}$ performance.

### 4.5 ROBUSTNESS TO DISTRIBUTION SHIFT

Previous work has shown ensembles help with metrics relating to uncertainty and calibration (Lakshminarayanan et al., 2017; Stickland & Murray, 2020). To assess this, we train JFT R50 HyperEnsembles and ExpertEnsembles for ImageNet classification. For the former, we use the BiT generalist; for the latter, we use the 244 experts, applying $k$NN to 50,000 examples from the training set to select experts. For both we use the validation set for greedy ensembling. Once the ensembles are trained and constructed, we assess them on a suite of datasets aiming to quantify robustness to distribution shift. Each dataset introduces some form of distribution shift (further details in Appendix B.2) - what we assess is the *accuracy* on these datasets. Figure 4 shows them. The expert ensembles offer a slightly better accuracy on the held out data; more importantly, they perform significantly better under distribution shift, improving over the HyperEnsembles by on average 2.2% across datasets.

## 5 RELATED WORK

We present literature related to the main aspects of this work. As well as previous highlighted novelties, we believe our contribution is distinguished from previous ensembling works by focusing on diverse datasets with production-scale deep models (instead of demonstrative smaller architectures and datasets), limiting the training data available, and formally assessing distribution shift.

**Transfer Learning**. Relating to a long history of research in manifold and representation learning (Tan et al., 2018), the transfer of features learnt by deep neural networks aims to reuse the abstract features learnt on a source (upstream) dataset in order to improve performance or data efficiency on a target (downstream) dataset. Bengio (2011) studied this in the context of unsupervised pre-training,

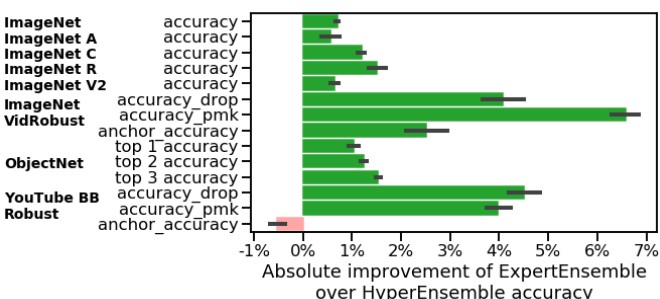

Figure 4: **Expert ensembles retain higher accuracy under domain shift**. Aside from the first bar, which shows test accuracy, all other bars correspond to some form of induced distribution shift, either artificially or otherwise. In all but one, we get significant boosts in accuracy compared to the HyperEnsembles.

proposing a number of ways to learn and re-use the features. Many works have shown benefits of transfer relating to convergence speed (Raghu et al., 2019), generalisation (Yosinski et al., 2014), accuracy (Zhai et al., 2019) and robustness (Djolonga et al., 2020), with the latter two showing particular benefits in the low-data regime.

**Ensemble Learning**. Known for improving models, ensembling methods have been studied in depth in and out of deep learning academia (Seni & Elder, 2010). There are few works which study ensembles in the context of transfer learning (Acharya et al., 2012; Sun et al., 2013). Bachman et al. (2014) pre-train entire ensembles on the source data and transfer, instead of transferring individual models. Work in the low-data regime is sparser. Using an ensemble of models from multiple training checkpoints, Laine & Aila (2017) label unlabelled data to then train individual models further, improving data efficiency for CIFAR100/SVHN. For few-shot classification on a new class, Dvornik et al. (2019) construct ensembles of mean-centroid classifiers from pre-trained ResNet18s.

**Deep Ensembles**. Lakshminarayanan et al. (2017) show that a simple approach of adversarially training multiple randomly initialised models from scratch and ensembling them yielded models with strong predictive uncertainty and calibration. Wenzel et al. (2020) showed that hyperensembles, which vary random initialisations *and* hyperparameters, outperform these deep ensembles.

**On Constructing Ensembles**. A key part of our algorithm is the use of the $k$NN to narrow down candidate pre-trained models into a relevant subset. Caruana et al. (2004) was arguably the seminal work studying how to select an optimal ensemble from a set of candidate models. A number of works extend AutoML frameworks (He et al., 2019) to explicitly optimise both the ensembling method and the members to maximise overall performance (Wistuba et al., 2017; Xavier-Júnior et al., 2018).

## 6 CONCLUSIONS

We have studied simple ways of creating performant ensembles with a limited amount of data. Overall, ensembles dramatically outperform their single-model counterparts. We show that diversity from upstream pre-training results in better ensembles than diversity induced downstream, regardless of whether this upstream diversity comes from pre-training multiple generalist models with different initialisations, using different architectures or specialisation via pre-training on different data. We demonstrate the efficacy of the nearest-neighbours classifier as an easily calculated discriminator between different pre-trained models, and even as a way to decide how many models to try on a downstream task, leading to convenient ways to combine both upstream and downstream diversity.

These ensembles achieve SOTA performance on the Visual Task Adaptation Benchmark at a significantly smaller inference cost, while also outperforming ensemble approaches relying on downstream diversity. They also exhibit higher robustness to domain shift as assessed by ImageNet variants.

There are many interesting avenues for future work. All our considered models were pre-trained in a supervised fashion, and this should certainly be extended to include other forms of pre-training. This approach of combining different pre-trained models is highly complimentary with efforts which train ensembles end-to-end with diversity-encouraging losses, such as those in Lee et al. (2015) and Webb et al. (2019). Lastly, works such as Batch Ensembles (Wen et al., 2020) and Parameter Superposition (Cheung et al., 2019) systematically decompose network parameters to compactly train ensembles. For pre-trained models with the same architecture, weights could be deconstructed to initialise those methods so as to benefit from transfer learning and make them feasible in the low-data regime.

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
