# OpenReview forum: "Deep Ensembles for Low-Data Transfer Learning"
_ICLR.cc/2021/Conference — Reject_

### Official Review · AnonReviewer4 · 2020-10-26
**Deep Ensembles for Low-Data Transfer Learning**

**Rating:** 5
**Confidence:** 4

**Review:**

Update after author response: I appreciate the authors' efforts to address my concerns and to raise some interesting points I missed. I still find the paper's insights are lacking some novelty to be published, but I think that this line of research is worth it!


------------------------


In the low data regime, the use of transfer learning techniques collides with a widely used strategy: training multiple models for different purposes, being the main obstacle the lack of a clear diversity source.

This paper proposes a simple method to circumvent this problem: identifying pre-training itself as an easily accessible and valuable form of diversity and proposing the greedy combination of several pre-trained models.  Experiments show that the proposed strategy can achieve state-of-the-art performance on 19 different tasks. One necessary assumption of the method is the availability of a large pool of related models and the ability to look at the target data to make a decision on which models to fine-tune.

One of the critical points of the method is the use of cheap proxy metrics which assess the suitability of a pre-trained model before training it.  To this end, the paper proposes the use of leave-one-out nearest-neighbour accuracy.

Pros:
+ The paper takes one of the most important issues of deep learning: training high-performance models in the low data regime.
+ The results section is well structured and experiments are convincing. The proposed method is evaluated from several points of view.

Cons:
- The paper refers to a previous publication (Puigcerver et al., 2020) and from this point of view, the proposal represents only an incremental step, with a low level of novelty.
- The description of the method is not very specific and it refers to other existing methods as the main steps (Puigcerver et al. (2020) and  Caruana et al. (2004))
- The proposed method is based on heuristics and there are no hints about why it does work. Diversity is a generic concept and there is a large number of papers that have explored several measures of diversity in order to understand "when" and "why" it is helpful. I miss some references to this previous knowledge. See, for example: Bian, Yijun, and Huanhuan Chen. "When does Diversity Help Generalization in Classification Ensembles?." arXiv (2019): arXiv-1910.

My main concern is not about the results, which I think are good, but about the level of novelty with respect to some existing publications (mainly Puigcerver et al. (2020)) and the lack of experiments devoted to understanding the role of diversity. It is a well-known fact that diversity per se is not sufficient to build strong multiple classifiers, and different kinds of diversity measures are helpful to diagnose it.

---

> ### Author Response · Authors · 2020-11-17
> **Reply to review by AnonReviewer4**
>
> We would like to thank the reviewer for the response - we are glad you recognise the importance of the work, and that the writing and experiments were of a sufficient standard!
>
> In response to some of the concerns:
> * __Novelty__
>     * On the topic of novel technical contributions, we agree with the reviewer that the work proposes an algorithm that builds off multiple preceding works, specifically the use of model selection to discriminate between pretrained models and the greedy algorithm for constructing ensembles. However, we argue that combining disparate efforts in order to present a computationally feasible and performant approach to tackle a problem of high importance is itself valuable progress.
>     * We also believe that our work has many contributions aside from the proposed algorithm:
>         * We found very few related works studying ensembling of modern neural networks in this data regime. We believe that studying different approaches is itself a solid contribution. For example, just demonstrating that the combination of transfer learning from a single pretrained initialisation + downstream diversity (from augmentations, hyperparameters, etc) can yield good performance in this type of task is itself a valuable finding, as we do not know of other works that show this in this data regime.
>             * For example, we know of only one other work that studies [hyperparameter ensembles](https://arxiv.org/pdf/2006.13570.pdf); they do not use production-scale architectures, do not restrict the data to the low data regime, and do not assess models on such a wide array of new tasks; and there is no transfer learning involved.
>         * Ultimately, we concluded that - whether the source of variation was due to upstream pretraining on different datasets, or simply upstream pretraining with multiple random seeds - that sources of upstream diversity are more performant than sources of downstream diversity. We do not find any concurrent works that demonstrate the same conclusion.
>             * We believe that the improvements to robustness demonstrated by the evaluation on ImageNet variants is a particularly interesting and valuable insight that makes this conclusion even more convincing, departing from the tunnel-vision focus on top-1 accuracy.
>             * Furthermore, demonstrating the superiority of combining the two sources of diversity is also a unique finding, and proposing a simple heuristic for doing so in a computationally efficient and performant way is a novel technical contribution.
>
> * __Why is it helpful - and what is the role of diversity?__
>     * Firstly, we think in hindsight that the term diversity was used very informally in this effort - it is tricky as there are multiple definitions and it isn’t something the field has agreed upon, but really when discussing upstream/downstream ‘diversity’, what we meant was sources of model variation as opposed to diversity in the sense of diversity of predictions/errors etc.
>     * That being said, we totally agree with the reviewer here - we read many papers on the topic of quantifying and understanding diversity and spent a lot of effort applying them to analyse our ensembles. Though we generally saw that the ensembles using upstream/combined diversity were more ‘diverse’, and that they were marginally more optimal on the diversity/accuracy tradeoff, we didn’t find any systematic and convincing trends, and didn’t wish to engage in data dredging till an attractive/presentable looking trend emerged. We also found this disappointing, as were hoping to find some clear insights to back up why the different sources of model variation help.
>     * The paper the reviewer linked is fascinating! It may explain the strong performance on the ImageNet variants. To our understanding, the role of diversity in ensemble performance still seems contested in literature, with many works proposing different diversity metrics and making contradictory claims about its importance; given this, and the fact that we already had a lot of strong results to present, we decided not to include it. We believe that in lieu of such analysis, thoroughly evaluating the different approaches from so many angles (19 diverse classification tasks for accuracy and 7 different variants of ImageNet for multiple robustness metrics) was a more useful contribution.

---

### Official Review · AnonReviewer3 · 2020-10-27
**Three main "contributions" in its framework are all from exisiting (related) works!**

**Rating:** 3
**Confidence:** 3

**Review:**

This paper does achieve good performance but its method is quite about engineering using very intuitive training tricks that everybody could be able to use given a lot of GPU machines. I would not like to encourage such work to be published as a research paper.

Pros:

1. The proposed framework achieves a good performance compared to its related works.

2. It is a good organization of a lot of training techniques, and a good reference for engineering.

Cons:

1. No technique contribution. The main framework of this submission is very similar to the existing work [Scalable Transfer Learning with Expert Models] which has not been officially published but only on arXiv. Besides the common methods of pre-training and ensembling, it involves three "new" methods in its main framework: the first one is kNN selection on pre-trained models (referred to the same technique in the work [Scalable Transfer Learning with Expert Models]); the second is the hyperensembles by fine-tuning multiple diverse copies of the models (referred to the hyperparameter sets used in another related work [Big transfer (BiT): General visual representation learning]); and the last is greedy ensemble (referred to the third related word [Ensemble selection from libraries of models]). Not sure what is the contribution of this submission.

2. The paper is quite about engineering tricks or combinations of tricks. In addition, in terms of engineering, it is not fair to compare to related methods under the condition of using the same numbers of pre-trained models. A better way may be based on the total computational COSTS such as the max running epochs, the network architectures, the total training time under the same usage of GPU machines.

---

> ### Author Response · Authors · 2020-11-17
> **Reply to review by AnonReviewer3**
>
> Many thanks for the comments; we hope we can address some of them here.
>
> 1. We understand the reviewers concerns relating to novelty in terms of technique contributions; we combine a number of disparate techniques (in particular, KNN selection from Puigcerver et al. and greedy ensembling from Caruana et al.) in order to build the proposed algorithm.
>
>     We would like to note a few technical contributions however:
>     * Previous work did not consider selecting multiple models for finetuning on new tasks (we also stress-test the KNN, picking only 15 models from a pool of over 2002!). Indeed, as far as we are aware, previous work found that combining different experts models trained on different data is hard, and it did not work out of the box. From Puigcerver et al.: “Selecting and combining multiple experts for any downstream task is a natural extension of our work. This could be especially useful for tasks that require understanding several concepts, not necessarily captured by a single expert.” In this work, we found a way to make it work nicely.
>     * We propose a heuristic based on KNN accuracy (Section 2.3.1) which allows a performant balance between upstream and downstream diversity by automating not only which pretrained models to select, but how many to select for each downstream task.
>
>     _However, aside from technical novelty, we believe there are a number of valuable contributions from this work:_
>     * To the best of our knowledge, we found no other works comparing or suggesting approaches to building ensembles of modern deep networks in the low data regime. We would like to re-iterate that there are only 1000 datapoints available per task downstream; for example, [concurrent work](https://openreview.net/pdf?id=_77KiX2VIEg) which developed ensembles in the low data regime achieved ~20% accuracy on CIFAR100, whereas our models achieve ~70+% accuracy (with only 10 data points per class). We further believe that our work is distinguished in considering production-scale models on a very diverse range of tasks, instead of experimenting with small-scale models on tasks such as MNIST. Furthermore, we find very little comparable work studying transfer learning and the low data regime in the context of ensembles.
>         * We note that even the performance of our baseline ensembles - which use ‘downstream’ diversity - are a contribution. Though we consider it a baseline, we know very few other works that study such ensembling techniques with such little data. Showing the competitiveness of hyperparameter or augmentation ensembles in this setting is in and of itself a contribution.
>     * Our main conclusion is that **upstream diversity is more useful than downstream diversity**; we know of no other concurrent or previous works that demonstrate this. Similarly, our demonstration of the superiority of combining both forms of diversity - and a proposed heuristic to do so efficiently - is also novel as far as we are aware.
>     * We believe that the results showing significantly increased distribution to domain shift, by assessing on the 7 ImageNet variants, are also a valuable contribution and a very interesting insight. Even if our models were not more accurate, we believe that showing such a significant boost on multiple robustness metrics is a very compelling result.
>
> 2. Concerning how models are compared:
>     * We are not sure we fully follow the point relating to fairness of comparison. We chose to compare inference cost, as this was the clearest thing that was comparable across settings. We do not make any claims in relation to the total training time - though given that, for example, the R152x4 requires ~45x as many FLOPs for a forward pass as a single R50x1, we suspect they are at least in the same ballpark range for training cost.
>     The core assumption of this work is that there is a number of available pre-trained upstream models, something which is increasingly true (see https://www.tensorflow.org/hub). In this setup, the downstream training cost of our ensemble methods is negligible, and most likely not the reason why we see the nice large gaps shown in Figure 2 a).

---

### Official Review · AnonReviewer2 · 2020-10-29
**Papers needs more clarity, better writing and total computational cost justification.**

**Rating:** 3
**Confidence:** 4

**Review:**

Summary:
Paper proposed an ensemble learning approach for the low-data regime. Paper uses various sources of diversity - pre-training, fine-tuning and combined to create ensembles. It then uses nearest-neighbor accuracy to rank pre-trained models, fine-tune the best ones with a small hyper-parameter sweep, and greedily construct an ensemble to minimize validation cross-entropy. Paper claims to achieve state-of-the art performance with much lower inference budget.

Recommendation: Based on my understanding of the paper I recommend a clear rejection. Please look at the details below:

Strength:
1) Authors have tried to lot of experiments and give summary of conclusion/results in section 4.

2) Experimental setup is clear and the motivation is valid.

Weakness/Questions:
1) Paper was very hard to read. I had to go back and forth between pages to make sense of what’s defined and make my own definitions in many cases. In some cases, terms are defined but never used and in other cases terns are never defined. For example,
a) AugEnsembles: Where is this used?
b) ExpertEnsembles: Where is this defined?
c) HyperExperts: Where is this defined?
d) AugExperts: Where is this defined?

2) In figure 2, Single-model SOTA has only one model. Do you have a graph for total cost (training + inference) vs VTAB_{1K} performance for all the models that are shown in figure 2? Only showing an inference budget may not tell the entire picture here.

3) In Table 2, how is computational cost different for different sources of diversity (D, U and C)? If C needs more computational cost than U and D then is the comparison fair?

4) Appendix A.2 mentions the hyper parameters used when using “hyper ensembles” and then there is a default hyper parameter sweep - “Default Hyper Parameter Sweep” in appendix A.1.
Did you find any pattern in the hyperparameters with the best model?  How were the hyperparameters chosen for baselines in table 1?


minor:
1) VTAB should have been defined just before listing contributions -
“new form of diversity improves on the Visual Task Adaptation Benchmark (VTAB) SOTA by 1.8% (Zhai et al., 2019).

2) Paper repeatedly cites Puigcerver et al 2020 [1] to justify experimental framework or as a follow up paper which is also very similar to the current paper in terms of motivation.

[1] Puigcerver, Joan, Carlos Riquelme, Basil Mustafa, Cedric Renggli, André Susano Pinto, Sylvain Gelly, Daniel Keysers, and Neil Houlsby. "Scalable transfer learning with expert models." arXiv preprint arXiv:2009.13239 (2020).

---

> ### Author Response · Authors · 2020-11-17
> **Reply to review by AnonReviewer2**
>
> Many thanks for the clear review! We will correct some of the minor formatting errors.
> In response to the comments:
>
> 1. Apologies for the lack of clarity in writing! We will definitely clear this up in the paper, but to clarify here:
>       * {Aug/Hyper} Ensembles are ensembles which utilise only downstream diversity (from augmentations/hyperparameter variation in the downstream finetuning).
>       * {Generalist/Expert} Ensembles are ensembles which utilise only upstream diversity (from pretraining models with multiple random seeds, or on different upstream datasets, respectively).
>       * {Aug/Hyper}{Generalists/Experts} combine both - e.g. HyperExperts utilise experts as an upstream source of diversity, combined with hyper parameterization as a downstream source of diversity. The proposed thresholding heuristic automatically decides the balance between the two forms of diversity.
>
> 2. This is a good point! We would like to mention a few things:
>       * The assumption underpinning many works in Transfer Learning is that the pretraining cost is only incurred once, and therefore not considered in the finetuning phase. We therefore split computation costs in ‘upstream pretraining’ (ignored - analogous to practitioners downloading pretrained models from the web), ‘downstream finetuning’ (consistent between all our ensembles), and ‘inference’ (arguably the most important)
>       * Assumedly, one needs to separately compare finetuning cost vs. VTAB-1K score, and inference cost vs VTAB-1K score, as a model is only trained once but may be used many times.
>         * Inference cost: We predominately compared based on inference budget as this info is readily available (we can use FLOPS as a proxy, which is likely fair due to the similarity in architecture). We believe inference cost is the fairer comparison given the differences in setup, and also of more practical relevance, and hope that the benefits in this quarter are clear in the paper.
>         * Fine-tuning cost: Here it is harder to say; these numbers are not readily available to us for the baseline, but the single-SOTA models were also fine-tuned downstream with a hyper-parameter sweep.
>             * As a back of the envelope calculation, the best single model is a R152x4 which has ~44x flops for a forward pass vs a single R50x1. Our ensembles train 60 ResNet-50s downstream. Assuming a training pass scales linearly in compute time w.r.t. an inference pass (a very conservative assumption), this would perhaps put our ensembles at 1.36x the cost of fine-tuning once the strongest single model we compare against.
>             * We showed (Section 4.4) that our methods perform very well even with reduced fine-tuning budget; our ensembles that train 20 ResNet-50s downstream still beat the R152x4 baseline. Thus we suspect our ensembles compare favourably here too.
>
> 3. As mentioned at the start of Section 4, all of our ensembles use the same finetuning/inference compute budget - Downstream, Upstream and Combined diversity are therefore on a fair playing field in this respect - we will adjust the text to make this clearer. The only difference lies in which (models+hyperparameters) we fine-tune in each case, but compute-wise they all get the same budget. One of the contributions of this work is to suggest a simple heuristic based on KNN accuracy which allows one to combine the two sources of diversity under a fixed finetuning budget.
>     * This again assumes the pretraining is ‘free’ - from a practical perspective, this is not an unreasonable assumption; there are many widely available pretrained checkpoints using a variety of architectures, pre-training methods, datasets etc.
>
> 4. Interesting question!
>     * The default hyperparameter sweep (2 learning rates, 2 learning schedules) is used for ensembling approaches (augmentation, upstream diversity) which don’t include a hyperparameter sweep already (e.g. AugEnsembles, ExpertEnsembles). It is in fact the same hyperparameter sweep used in the original Visual Task Adaptation Benchmark paper. Concerning patterns between these hyperparameters and the wider sweep used to generate HyperEnsembles, we didn’t notice any systematic trends between the hyperparameters across the 19 tasks; it was highly task dependent, with per-task preferences relating to dropout, learning rate, schedule length etc varying significantly.
>     * The baseline numbers are copied from literature and were not replicated by us. They propose a hyperparameter heuristic which adaptively sets learning rates/schedule lengths/resolution/etc as a function of the downstream task’s properties (number of images, nature of the images and so on); downstream, they therefore only use one hyperparameter per task. However, this hyperparameter heuristic was defined after much experimentation on VTAB-1k, whereas we use the default suggested in the original Visual Task Adaptation Benchmark paper; therefore, it’s not simple to compare the ours to the baseline on this front.

---

### Official Review · AnonReviewer1 · 2020-11-01
**Recommendation to Reject based on limited novelty and lack of convincing experiments**

**Rating:** 4
**Confidence:** 5

**Review:**

[Summary] This paper presents different ways of creating ensembles from pre-trained models. Specifically, authors first utilize nearest-neighbor accuracy to to rank pre-trained models, then fine-tune the best ones with a small hyperparameter sweep, and finally greedily construct an ensemble to minimize validation cross-entropy. Experiments on the Visual Task Adaptation Benchmark show the efficacy of the approach in selecting few models within a computational budget.

[Score] Overall, I found the paper is well-written with experiments using large-scale benchmarks such as JFT, ImageNet21K and VTAB datasets. I like the problem of model selection for transfer learning. However, my major concern is about the novelty of the paper including concerns regarding prior works. Given the lack of novelty and convincing experiments, I vote for rejecting the paper. Hopefully the authors can address my concerns in the rebuttal period.

[Weaknesses] The technical novelty of the paper is very limited. Besides combining few prior methods (e.g., Puigcerver et al. (2020); Caruana et al. (2004)) and then performing large scale experiments on JFT/ImageNet21K datasets, what are the main contributions of the paper are not clear. Although I admit that papers on analysis or study of different methods are quite interesting, I failed to find any major insights from the study of different diverse ensemble techniques. Is the upstream pre-training achieves better accuracy than that from the downstream fine-tuning stage the major take away message of the paper? Authors should clearly explain the major contributions of the paper.

There are few recent papers which discuss model selection for transfer learning. E.g., Duality Diagram Similarity: a generic framework for initialization selection in task transfer learning, ECCV 2020; DEPARA: Deep Attribution Graph for Deep Knowledge Transferability, CVPR 2020. How is the proposed approach related to these prior works? These paper should be clearly discussed with proper comparison in the experiments.

Comparison with prior methods is not satisfactory. Authors should clearly discuss what are the different ways of selecting models and creating ensembles out of that in the experiments. Specifically, what are the different alternatives to KNN and greedy approach used to construct ensembles? What about the performance of those methods? How is the proposed simple approach comparable to them in terms of performance vs complexity. E.g., how is the proposed approach comparable to the pretrained model selection strategy based on Task2Vec: see TASK2VEC: Task Embedding for Meta-Learning?

How is the proposed method related to Leep: A new measure to evaluate transferability of learned representations? Furthermore, how is the current approach comparable to a simple baseline on fine-tuning with early stopping?

Figure 1 is not clear and it is not described clearly anywhere in the paper. I would like the authors to clearly explain this figure either in the caption or text in the introduction section.

---

> ### Author Response · Authors · 2020-11-17
> **Reply to review by AnonReviewer1**
>
> We thank the reviewer for the thorough response! We are glad you enjoyed the writing and found the explanation/experiments of a good standard, and hope we can address your comments below:
>
> * About novelty of the work: We do believe this paper has many valuable insights, and will aim to make that clearer in the updated version.
>   * We would first like to note that as far as we could find, there were very few research efforts focussed on ensembles of modern neural networks in the low data regime - we reiterate that on downstream tasks, we only have 1000 data points available. For tasks such as CIFAR100 or CalTech101 this means only 10 data points per class. We believe it is a solid contribution to demonstrate the efficacy of different ensembling methods (both utilising ‘downstream’ sources of diversity and ‘upstream’ sources of diversity). For context, [recent work](https://openreview.net/pdf?id=_77KiX2VIEg) studies ensembling in a similar regime - their results on CIFAR100 (10-shot per class i.e. ~1k data points) achieve around ~20% top-1 accuracy, but our approaches achieve 70%+. Considering production-scale models, on realistic and diverse classification tasks, with very small amounts of data, is a very useful regime to work on and we believe there are many useful insights here for practitioners and researchers alike.
>      * We recognise the kNN-score for model selection component of the algorithm isn’t a novel contribution, but believe that the application to selecting a set of models, to be combined later, and stress testing it (picking 15 from over 2000 models!) are highly valuable contributions.
>     * We further note that the simple yet highly performant heuristic for combining upstream and downstream diversity is also a novel technical contribution.
>   * As you noted, yes; one of the key messages of this paper is that in this regime, creating ensembles which leverage differences in pretraining perform better than models which exploit diversity that is generated on the downstream task. As far as we are aware, there is no other concurrent work which demonstrates this conclusion. We also propose heuristics for computationally efficient ways to get the best of both worlds - again, we do not find parallels in the literature.
>   * Departing from the low-data regime and top-1 accuracy, the robustness on ImageNet results are arguably even more important than raw top-1 accuracy in a controlled setting, and we believe it is a valuable contribution to show ways to tackle it.
> * Points about related literature:
>   * Many thanks for pointing out some of these other papers! We will update our work to properly compare and contrast against these works.
>   * As far as we understand, LEEP, DEPARA, Dual Diagram Similarity, and Task2Vec all propose different ways of selecting models - but they all focus on selecting a single model for a given task. Those methods could all act as a drop-in replacement for the KNN selection phase of the proposed algorithm, and are thus complementary to our work - for example, if using LEEP in the model selection phase improved ensemble performance on downstream tasks, then we believe that this would only further verify the efficacy/performance of our approach, as opposed to being a literature baseline that we did not evaluate against.
>   * The performance of our approach does not directly necessitate the use of the KNN / the ‘pre-selection’ phase; we feel it was a valuable contribution to show that these sorts of approaches can help narrow down the pool of potential models, thus making the algorithm computationally feasible for many practitioners. However, one of our best results was just using a pool consisting of models pretrained upstream with different random seeds. This is analogous to practitioners using multiple different pretrained models (e.g. on ImageNet), which are widely available online.
>     * Lastly, the main findings and contributions of our paper - in relation to the superiority of upstream/combined diversity instead of downstream diversity, the impact on robustness to distribution shift and the proposed algorithm itself are all novel with respect to the papers mentioned.
> * Comparing with simple baseline of fine-tuning with early stopping: This is effectively what the ‘single model’ baselines are. Previous efforts fine-tune a single model on each downstream task, and found that such a strategy applied to very large scale models was the most performant approach. We achieve higher performance using smaller models with a significantly lower inference time. Note that those models do not require early stopping as after significant study on VTAB they developed a hyperparameter heuristic rule which defines schedule length as a function of the downstream dataset; this is arguably an even stronger baseline than finetuning with early stopping. Furthermore, early stopping could be applied to the baseline as well as our ensemble models, and therefore we consider that a separate direction.

---

### Decision · Program_Chairs · 2021-01-07
**Final Decision**

**Decision:**

Reject

**Comment:**

All reviewers recommend that the paper be rejected.  The reviewers appreciate the line of research and is worthwhile, but find that the paper lacks in technical novelty and insight.  The AC is in consensus with their reviews due to the concerns raised regarding novelty and insight and recommends rejection.